# New Approach for Extrusion Additive Manufacturing of Soft and Elastic Articles from Liquid-PVC-Based Consumable Materials

**DOI:** 10.3390/polym14214683

**Published:** 2022-11-02

**Authors:** Bohdan Savchenko, Nadiya Sova, Victor Beloshenko, Bohdan Debeluy, Aleksander Slieptsov, Iurii Vozniak

**Affiliations:** 1Department of Applied Ecology, Technology of Polymers and Chemical Fibers, Kyiv National University of Technologies and Design, NemirovichaDanchenko Street, 2, 01011 Kyiv, Ukraine; 2Donetsk Institute for Physics and Engineering Named after O.O. Galkin, National Academy of Sciences of Ukraine, Pr. Nauki 46, 02000 Kyiv, Ukraine; 3Centre of Molecular and Macromolecular Studies, Polish Academy of Sciences, Sienkiewicza Street, 112, 90001 Lodz, Poland

**Keywords:** additive manufacturing, liquid consumable, polyvinyl chloride plastisol, mechanical properties, soft and ductile articles

## Abstract

The article deals with the experimental development of a novel additive manufacturing (AM) process using a liquid consumable based on polyvinyl chloride plastisol. A conventional additive manufacturing system designed for deposition of melt filaments was converted to deposition of liquid material. Additive manufacturing with liquid plastisol enables the production of parts with low Shore A hardness and high ductility, surpassing the performance of the conventional filament process. The novel AM process enables the production of articles with a Shore A hardness of 5 to 60, and the mechanical properties of the additively manufactured articles are similar to those produced in the mold. This was achieved by varying the parameters of the AM process as well as the composition of the plastisol composition, including those filled with an inorganic filler. The application of different material distribution patterns also has a significant effect on the mechanical properties of the samples. A potential application of the investigated AM method was proposed and practically evaluated.

## 1. Introduction

Additive manufacturing (AM) is an emerging technology for producing solid objects from a layer-by-layer digital design [1]. As an important part of the fourth industrial revolution, AM is penetrating almost all available industrial markets and segments [2]. The main potential and advantage of AM is the ability to produce small quantities of complex and customized items with constant manufacturing costs [3,4].

Today, AM technology covers a wide range of materials, including polymers, metals, ceramics, traditional building materials, food, and many others. What recently began as rapid manufacturing is now a complete manufacturing technology that bypasses traditional production laws and principles [5,6,7,8]. Among many basic materials, the application of polymers for AM is the most widespread AM segment in the world in terms of the number of AM devices already produced. Polymer AM is represented by various consumable material forms—liquid resin, filament, powder, and granules—and is used in a variety of AM methods [9,10,11].

Extrusion AM is the method of depositing material that exits a spinneret or capillary. The material can be used in a liquid or molten state, depending on the initial conditions [12,13]. The most important process is material extrusion, and specifically melt extrusion and its most widely used type-fused filament fabrication (FFF) [14], where the initial material is used in the form of a solid filament. The main advantage of FFF is the availability of common parts and the simplicity of equipment design. Newer, not even household, FFF processes can produce industrial-grade and fully functional parts using high-quality engineering materials.

Conventional FFF processes use filaments to feed material into the melt nozzle and material properties are limited by the consistency and stiffness of the filaments to feed successfully [15]. Filament feeding is a critical step for the stability of the extrusion process, and the key factor for this is the axial force generated by the roll feed mechanism. Rigid materials with a high flexural modulus can be easily processed due to the high axial force that the filament exerts on the molten material in the extruder. Elastic and soft materials can be bent in the transition cavities before they reach the extruder assembly, creating significant drag forces that reduce axial force and feed rate stability.

The extruder for the FFF AM process is a simple piston design in which the material, in the form of rods or filaments, is both the piston and the material feed. The material fed through two or more drive rollers must be stiff enough to exert some linear force and resulting pressure in the extruder die. Soft materials that deform both plastically and elastically generate very high wall pressure forces that cannot be realized withdrive rolls [16]. Only stereolithography and resin extrusion processes can use soft and elastic resin materials with high detail and resolution [17].

Conventional AM processes use soft and elastic materials in the various technological AM processes. Photopolymer-based lithographic AM uses a UV-light-curable resin composition with high elasticity and softness [18]. The main advantage of this process is the extremely high resolution and the wide range of materials; the limitation is the complex post-processing and the problems with the resin. Extrusion of water-curable silicone monomer by material extrusion [19] is the most developed process for elastic material AM. The main advantage of this process is the remarkable complexity of the properties of silicone elastomers and the main disadvantage is the limited curing speed of the currently available silicone chemistry. The AM process for selective sintering or melting of powdered elastomer materials is a well-established process for the production of elastic articles [20]. The cost of equipment and material is very high and the choice of materials is currently limited to thermoplastic polyurethane and some other elastomers [21]. Screw extrusion AM is another option for the application of elastomers in the material extrusion AM process [22]. Almost all elastomers that are in flowable form can be processed by the screw extrusion process. The screw extrusion process has great potential, but is currently not widely used in the industrial AM segment.

Polyvinyl chloride (PVC) is a very versatile polymeric material with exceptional properties. Almost any object, from very hard to extremely soft, can be manufactured from the composition based on this polymer. The main disadvantage of PVC is its low thermal stability and low environmental and recycling compatibility. Moreover, the high chlorine content makes this polymer dangerous when incinerated and contaminated by other recycling streams [23,24].

Strong intermolecular forces can be weakened by suitable plasticizers, which transform the naturally hard PVC material into an elastomer with valuable properties. Depending on the plasticizer content and type, the PVC compound can be converted in powder, paste or liquid form [25,26].

Plastisol is a liquid suspension of PVC solid particles coated with a liquid shell containing plasticizers. The most important property of plastisol is its viscosity, which is highly dependent on temperature. Sorption and migration of plasticizer from the surface to the particle core significantly affect the interparticle distance and mobility. When the system temperature is increased, the liquid PVC suspension can change to a viscous, pasty state then to the gel and melt state, and finally, after cooling, to the solid plasticized PVC. Depending on its composition, liquid PVC plastisol usually contains a high proportion of plasticizers and, after heat treatment, becomes a flexible PVC with a low Shore A hardness index and low tensile modulus [27,28].

The typical Shore A hardness for plastisol-derived flexible PVC is 30 and lower. Flexible PVC material with low Shore hardness is an elastic material with typical elastomeric properties, which is very valuable for some applications.

The conventional FFF AM process, which uses a filament feed, is not suitable for low Shore hardness materials because of the feeding blockage. The special design of the FFF extrusion system, known as direct drive extrusion, can improve the supply of elastic material. The Shore hardness A70–A80 is the typical limit for the conventional filament-fed AM [29,30].

The aimof the current study is to apply the temperature-dependent behavior of plasticized PVC to the extrusion AM process. This idea has been experimentally confirmed by numerous trials and equipment designs. Application of unique physical properties of PVC in the AM process allows to enhance the range of available materials and property options.

The main objective of the current study is to develop an AM production process for soft and ductile articles with application of simple and widely available hardware based onthe FFF AM process and utilize low-cost and readily available consumable material. Another goal is to develop an AM process with the possibility of fast and easy incorporation of various organic and inorganic additives and the possibility of extension to a multi-material and high-flow AM process. The developed AM process is aimed at practical application in the field of production of programmable foam structures with absorption properties and volume permeability for gasses and vapors.

## 2. Materials and Methods

### 2.1. Materials

The PVC resin used was the Vinnolit EP 6854 emulsion type with a Fikentscher’s Constant value of 67. The plasticizers used in the composition can be biologically or synthetically based. Epoxy-modified soybean oil (ESO) and diisononyl phthalate (DINP) were used as plasticizers. Barium zinc stearate complex and calcium octoate were used as stabilizers. Omyafiber 800 ^TM^, a high-quality calcium carbonate-based filler with an organic coating was supplied by Omya AG Ukraine representative (Kyiv, Ukraine).

Typical plastisol compositions used in the study are listed in Table 1, where the component content is given in parts per hundred in relation to the PVC content. PVC resin in powder form was used after sieving on a vibrating sieve with a sieve opening of 100 microns to remove large particles. Mixing of the resin with the plasticizer was performed in two steps. The first mixing was conducted in a vertical batch dissolver (IKA RW20) for 6 h. The second mixing was conducted in a three-roll mill (Exact 35 with ceramic rolls) with two passes through the mill at the minimum roll gap setting. The mixed plastisol was stored in a plastic container and mixed in this container prior to the AM process. Alternatively, an open mixer with double Z-blades and water-cooling jacket was used as a single-stage mixer with different duration and rotation speed.

### 2.2. AM Experimental Setup

The most common design of AM hardware (3d printer) was used as a starting point for the experimental work. Figure 1 show the standard AM setup with an FFF extrusion system based on the filament used as a starting point.

In general, the AM setup consists of a deposition table, an extrusion system, and an axis movement system and an electronic control system. (Figure 1a). The most specific part is the material feeding system (Figure 1b), which consists of a mechanical roll feeder, an extruder, and a cooling device. Filament extrusion uses a ram extrusion system in which the filament is forced into an endless ram by a roller mechanism. The ram extruder melts the material in a laminar flow without mixing or homogenizing the melt, and the thermal conductivity of the material and the axial force of the ram affect the overall throughput of the process. A low internal volume and a short residual time allow the use of high processing temperatures compared with traditional screw extrusion. The extrusion line consists of two main stages—the cold stage and the hot stage. The cold stage or thermal barrier is designed for mechanical connection of the hot stage to the machine body. The hot stage is a malting chamber containing a heated block with a replaceable die. Melting of the material takes place in the nozzle body, which is surrounded by a heating block with heating elements and a temperature sensor.

During the tests, it was found that the standard extrusion line can be used for the extrusion of liquid material without any modifications. Only a mechanical seal with a high-temperature sealant is required for leak-proof operation.

The design widely known as Prusa I3 [31], in which the deposition platform moves in the Y direction and the extrusion system moves in the X and Z directions, was converted from filament feeding to liquid feeding.

The feeding system was converted by replacement of the filament roller feeder with a liquid feeder by using a stepper motor-controlled dosing pump Figure 2.

The control signal from the AM device motherboard was routed to an external stepper motor driver and then to the stepper motor.

The AM unit was enhanced with a quick-change device for rapidly changing extrusion dies from filament to plastisol feed, allowing both types of material to be used in the same unit. The AM build volume was 300 mm × 300 mm × 300 mm.

Various types of pumps were used for testing, but the most suitable found was asmall gear pump with a displacement of 0.15 and 0.3 mL^3^ per revolution. The gear pump was equipped with an adapter that allows the connection of standard pneumatic fittings with 1/8-inch thread. The gear pump shaft was connected to a standard two-pole geared stepper motor with a reduction ratio of 1 × 23. The gear pump motor was connected to a separate driver and controlled from the main board using step commands from G-code files. The extrusion speed was controlled by setting the extrusion multiplier, filament, and die diameter in the standard slicer software after recalculating the volumetric flow rate and material density.

The extrusion setup was used as a standard for the filament process with a water cooling thermal barrier. During the experiment, it was found that a water-cooled system was preferred and had higher performance. Therefore, a heat block with water cooling was used for all samples in this study. An air-cooled heat block can also be used, but at a slower feed rate. Successful operation requires a very fast temperature transition between the extruded heat block and the heat barrier to prevent clogging of the material.

The heated build platform was used without modification. Different types of coatings for the build platform were tested and evaluated to achieve adhesion of the parts. Polyvinylpyrrolidone, polyvinyl acetate, polyvinyl butyrate, and PVC polymer solutions were used. Polyvinyl acetate and PVC solutions are the most suitable. Figure 3 shows a picture of the current AM setup.

The most important process parameter is the temperature of the heated extrusion block, which was selected based on manual tests. The extrusion feed rate was calculated based on the volumetric feed rate and the characteristic volume of the gear pump and applied in the slicer software based on the filament diameter and extrusion coefficient parameters.

The conversion of the 3D model into the machine code was carried outusing the standard slicer software UltimakerCura 4.1 with manual recalculation of the feed rate.

Liquid plastisol material can be converted to flexible PVC by heating and cooling. This simple pathway consists of a chain of complex physical processes—plasticization, gel formation, melt formation, and melt cooling. The final product is a solid plasticized PVC—a thermoplastic polymer.

The main approach of the study—to complete all physical transformations of PVC plastisol—is the AM process to produce soft and elastic articles.

The plastisol composition for the AM process should have a certain viscosity value in order to be pumped by the selected pump type. The gear pump used in the study requires a maximum viscosity of 2–3 Pas to be pumped at hydrostatic pressure. In order to increase the maximum viscosity, a pressure chamber was used that allows plastisol with a viscosity of up to 15 Pas to be used.

Heating of PVC plastisol is accompanied by a sharp increase in viscosity. The pump pressure should be sufficient to overcome this pressure increase so that the molten material can be extruded from the die.

In the extruder system, the plastisol is rapidly heated in the heating block. During this process, the PVC suspension is converted into a viscous gel and then goes directly into the melt state. All this should be completed in a short time and with low volume to achieve low pump pressure and low residence time in the molten state. The AM experimental setup was equipped with an electronic pressure sensor in the output line of the gear pump. The material pressure was measured during extrusion at different flow rates and temperatures.

The AM process is generally highly dependent on the type and production technology of the consumables. The most widely used process is FFF AM due to the availability of filament consumables. The use of filaments in AM is simple, but the overall process including filament production from raw material is quite complex.

The production of consumables for FFF AM is quite complex and multi-step, requiring specialized equipment (Figure 4).

In comparison, consumables based on liquid PVC plastisol can be processed in a simpler way using conventional equipment (Figure 5).

In the case of the introduction of different additives, the PVC plastisol process does not require additional equipment for the distribution of additives, while the filament process in this case requires a compounding step.

### 2.3. Characterization

Mechanical properties—tensile strength and elongation at break—were measured on standardized specimens according to ISO 527-2:2012. Shore A hardness was measured on the same specimens with a thickness of 10 mm ISO 7619-1:2010. Material density was measured by hydrostatic weighing in a water–alcohol mixture ISO 1183-1:2019. Melt flow rate was measured according to ISO 1133 with a capillary of 2.095 mm diameter and a weight of 2.16 kg.

The viscosity of the liquid plastisol was measured from room to process temperature using a Brookfield Cap 2000+ rotational viscometer, the heating rate was 100 degrees per minute, and the rotation speed was 50 rpm. Temperature programming was performed using a PC-basedinstrument control software.

## 3. Results and Discussion

### 3.1. Material and Process Characterization

The first part of the experiment was to determine the successful process parameters for stable and repeated performance of the AM process with liquid plastisol. Viscosity-temperature data for non-filled samples (numbers 1–4, Table 1) was used for the evaluation of a possible processing window as shown in Figure 6.

From the data shown in Figure 6, it can be seen that the gelation of plastisol starts at about 100 °C as a moderate viscosity increases and turns into melting above 130 °C. At 140 °C, melting can be seen as a sustained viscosity increase. After melting, the viscosity of the material decreases with increasing temperature. This temperature range is most suitable for the AM process. As the temperature continues to rise, the viscosity increases due to degradation and crosslinking.

Liquid PVC plastisol was converted to solid flexible PVC by heating in a flat mold that had the shape of the test samples. The properties of the flexible PVC samples prepared from the plastisol compositions used in the study are shown in Table 2.

As shown in Table 2, the DINP plasticizer is more effective than ESO at low dosage (100 pph) composition with 100 pph DINP having lower softness than with 100 pph ESO; however, at high dosage (200 pph) sedimentation of the composition occurs. The compositions with ESO have higher thermal stability due to the stabilizing effect of the epoxy groups [32]. Based on the rheological data, a temperature range of 170 to 210 °C was chosen for the practical experiments.

For comparison, the properties of molded samples were compared with AM samples. Liquid plastisol was poured into a mold and converted into flexible PVC material by heating and cooling. Heating temperature was 190 °C at a duration time of 5 min followed by cooling in a closed chamber for 30 min. The AM process parameters were introduced during the slicing of the test models and are listed in Table 3.

Extrusion temperature and feed rate are the main process parameters responsible for the successful performance of the AM process. To successfully convert the sol into a melt, the material must be treated with temperature and pressure (called the plasticization process).

In order to determine the correct process parameters, static experiments were performed on an AM unit with a working extrusion unit and no axial movement. The extruded material was fed by means of a gear pump with different flow rates, temperatures, and time. Because of the thermal degradation of PVC, the feed rate can be lowered as well as the maximum possible melting time in the extruder.

Extruder inlet pressure was used as a response parameter for static extrusion tests. Stable pressure values over time were identified as stable extrusion conditions, as well as extrudate shape and flow stability. A temperature range of 180 to 195 °C was selected for these experiments to ensure stable pressure over time and stable flow of the molten polymer. A feed rate of 150 to 800 mm^3^ per minute is possible at an extrusion pressure of 10 to 40 bar. A feed rate lower than 150 mm^3^/min is possible with sufficient thermal stabilization because of the long residual time in the test setup. The degradation of plastisol can be seen from the increase in pressure over time and the discoloration of the extruded material. The present process parameter setups (Table 3) were selected because they can provide stable pressure values over a longer period of time—2 h. During the experimental work, it was found that the feed rate for plastisol can reach 1000–2000 mm^3^/min, which is higher than the typical feed rate for the 1.75 mm filament feeding process. When using an extruder die with a long residual time, an even higher throughput is possible. The main limiting factor was the hydrostatic pressure resistance of the pipes between the extruder assembly and the gear pump.

### 3.2. AM Material Characterization

Samples with 100% filler were 3D printed with different compositions and process parameters to study the mechanical properties and hardness. The AM specimens with monolithic structure were prepared by adjusting the extrusion speed and measuring the weight-to-mass ratio of the printed specimens.

Typical properties of AM specimens fabricated from PVC plastisol material are listed in Table 4.

The tensile strength and elongation at break of the AM specimens are generally lower than those of specimens made in molds. This is possibly due to the influence of cohesion between layers and internal voids. The Shore hardness value is generally lower than molded parts and is the same for filled specimens. Materials with high plasticizer content have a property complex close to that of the molded part, which may be due to lower melt viscosity and more effective cohesion during the AM process.

The properties of the samples with inorganic filler are very close to those of the unfilled samples with the investigated content. At high filler contents, sedimentation of thefiller was observed. If the filler content is high, special additives should be used. In general, the AM process can be successfully carried out with composite fillers and can be used, among others, for coloring or hardness adjustment.

As can be seen from the comparison of Table 1 and Table 3, the mechanical properties of the AM specimens are of the same order of magnitude as those of the molded specimens, and their properties may correspond to the final application.

### 3.3. Influence of AM Layer Orientation

Layer cohesion has a decisive influence on the mechanical properties of the AM material. The unique capability of extrusion AM is the programmable ability to orient layers and material streams within the part. The material streams can be positioned in different ways according to the direction of deformation. Figure 7 shows the arrangement of material streams used in the slicing software.

The influence of the alignment of the material layer is shown in Table 5. The highest tensile strength and elongation at break values are observed for the 90-degree current orientation and the lowest for the 0-degree orientation. This behavior is typical for AM specimens and can be explained by the drawing of the melt through the spinneret and the cohesive interactions between the material streams. For specimens with a layer orientation perpendicular to the direction of deformation, the tensile strength is greatly reduced because it is transferred through cohesive interfaces between the material streams. In general, the cohesion-based material bond is much less strong than the solid material steam. Compositions 1 and 2 and 5 and 6 were chosen in this experiment in order to evaluate filler influence on material layer distribution. The presence of inorganic filler reduced tensile parameters, and elongation at brake is the most sensitive parameter indicating filler presence. Property deterioration can be caused by cohesion blockage on the stream’s interface.

The unique feature of AM technology is the programmed transformation of the object into a core-shell structure, a shell structure, and an infill structure. This capability allows programming of material distribution in the volume of the AM part. Infill structures with non-monolithic volume filling create a material structure with air-filled voids in the interior. Programmable infill structures produced by the AM process can be represented as a foam structure or programmable foam structure. The main advantage of such structures is the possibility of reliable control of cells and wall structure, which is a challenge for traditional foam technology. The application of polymer foams and foam structures is widespread and well established. The application of different material properties in AM of foam structures can significantly affect their complex properties. Foams made from elastic polymers such as polyurethane, ethylene vinyl acetate, and rubber are a very important industrial family of materials. Insulation properties, shock and wave absorption, are the most important properties of such materials.

Infill structure was used in the AM process of PVC plastisol to change the material hardness. The influence of slicer settings of infill density on Shore hardness is shown in Table 6 as rectilinear infill pattern and gyroid pattern structures.

By applying the programmable filling density, it is possible to change the material hardness in a wide range. The influence of the filling pattern structure can be used not only for hardness control but also for directional property adjustment. The rectilinear structure is a two-dimensional structure, while the gyroid structure is a three-dimensional structure with uniform distribution of the material [33].

### 3.4. AM Process Specific Limitations

During the experimental work, some limitations of the plastisol AM process were found. Thermal decomposition of palletized PVC in the heated extruder parts may lead to uncontrolled viscosity increase and nozzle clogging. Plasticized PVC is not as susceptible to decomposition as rigid PVC, but if it is heated above the melting point for an extended period of time without the material flowing, decomposition will occur. Metal surfaces can catalyze the decomposition. Experimental tests have found that copper and its alloys are not suitable at temperatures above 190 °C PVC AM, as the metal surface is severely decomposed and further clogged. Stainless steel is much more suitable and can be used at higher temperatures.

In general, PVC plastisol material with DINP plasticizer and stabilizer complex can be heated at 190 °C for up to 10 min without flowing before the nozzle clogs. The composition with ESO is more stable and can have a lifetime of up to 20 min.

The composition without the stabilizing complex can be processed, but with low heating and no flowability. The composition with the ESO plasticizer is especially suitable for processing without the stabilizer. To overcome this limitation, some software improvements can be realized, such as the use of material conveying the flushing or fast cooling of the extruder unit.

Another process-limiting factor is sedimentation and separation of the plastisol. PVC plastisol must be mixed prior to the AM process so that it is stable during the AM process. During storage in the pressure chamber, it can affect the sedimentation process and separation into a two-component system. The sedimented component can block the flow in the pipes and gear pump. Experimental tests have found that suitable mixing conditions during plastisol preparation can generally eliminate this problem. The best mixing conditions are to use a three-roll mill in combination with an overhead mixer or to use an open mixer with two rotors. The mixing time is of great importance. When an overhead mixer and three-roller mixer are combined, the typical mixing time is 3 to 5 h in the overhead mixer and two to three passes in the trough mill. It is more favorable to use an open two-roll mixer with single-stage mixing with a duration of 4 to 5 h. The use of surface-active agents with a sedimentation-inhibiting effect is strongly recommended. Another technological step to reduce sedimentation is the mechanical screening of PVC resin with a screen size of 100 microns using a laboratory vibrating classifier.

### 3.5. Plastisol AM Process Applications

Possible applications of the developed AM technology are articles and materials with low Shore hardness and high ductility.

An open-cell material with shock- and vibration-absorbing properties and a Shore A hardness of 36 was successfully fabricated and used as a shock-absorbing layer in a body armor structure (Figure 8). The material has favorable shock distribution properties and water vapor permeability properties, which were achieved by using a gyroid filling structure. Various types of molded profile O-ring seals have been successfully fabricated under laboratory conditions and used in various applications.

Fishing lures with a Shore A hardness of 12 to 25 have been successfully tested in the laboratory and have proved their worth in fishing.

The developed AM process has been successfully applied to the fabrication of human body imitations. The 3D model was created based on a tomographic scan of a patient’s nasal cavities. The AM part, which is very similar to the human body due to its hardness and consistency properties, was used for surgical exercises before treating real patients. A material with a Shore A hardness of 18 to 25 and an infill density of 60 to 80 was used to achieve tactile properties similar to the human face. AM models allow clinicians to perform successful surgical treatment, which is used for training and educational purposes (Figure 9).

The training models were created in a natural scale with red organic die coloring. The PVC plastisol after the AM process has a unique surface appearance—well developed surface roughness which is a possible reason for the specific flow structure in the nozzle. The well-developed surface roughness reduces the surface gloss of the articles and creates a soft-touch effect on the article surface, giving the articles produced with plastisol AM a completely different appearance compared with the conventional FFF AM process.

## 4. Conclusions

The AM process, which uses liquid consumables, is being practically developed and studied. The newly developed technology enables the production of materials with extremely low Shore hardness, which is impossible in the current powder-based AM processes. The developed AM process can be easily realized by rebuilding the conventional FFF AM equipment with additional parts. Software and electronic parts can be used without modification. Consumables are readily available and can be manufactured on site.

The application of model fills in 3D-printed material makes it possible to further reduce the hardness of the material as well as the mechanical properties. The mechanical properties of AM material are lower than those of molded material, but can be improved with the right process parameters and additives. The mechanical properties of AM materials depend on the orientation of the layers, similar to the filament AM process.

Filler components can be introduced into the liquid plastisol, enabling an AM composite process with liquid consumables. Coloring of the AM material is easily achieved by using pigments or matrices.

The AM material can be used to make gaskets, shock-absorbing parts, fishing lures, and other products. AM parts made by this method have been tested as shock-absorbing structures for ballistic protection articles. AM with liquid plastisol is a promising technology for producing composite materials. Plastisol’s low viscosity and ease of processing and formulation make it an easy medium for incorporating various fillers and nanoparticles.

## Figures and Tables

**Figure 1 polymers-14-04683-f001:**
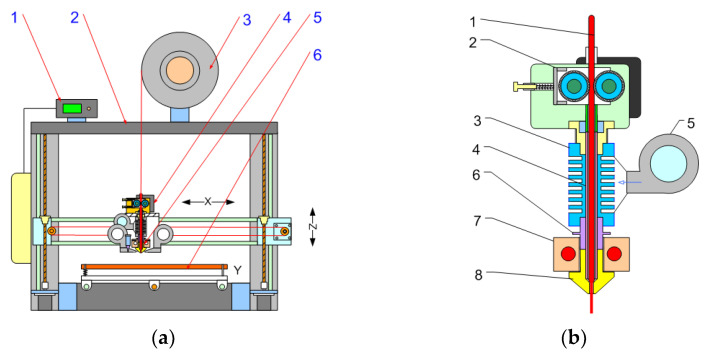
Conventional FFF AM setup: (**a**) general view: 1—electronic control system; 2—machine frame with axial drives; 3—filament spool; 4—roll feed; 5—extruder setup; 6 deposit platform; (**b**) extrusion system: 1—filament; 2—roll feed; 3—heat sink; 4—inner liner; 5—cooling fan; 6—thermal barrier; 7—heater block setup; and 8—nozzle.

**Figure 2 polymers-14-04683-f002:**
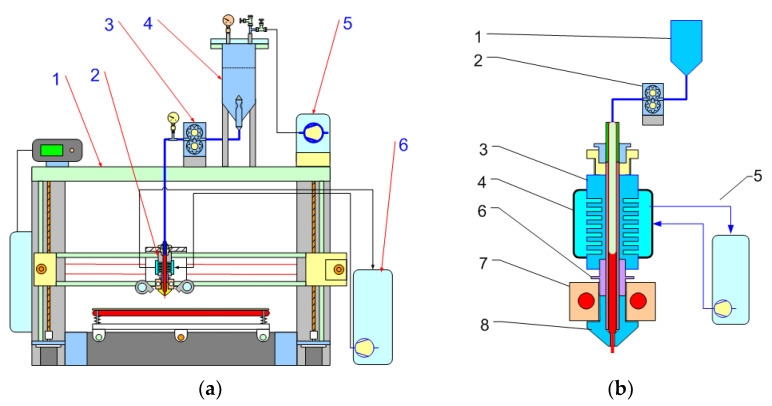
Experimental setup for AM with PVC plastisol: (**a**) general view: 1—Prusa I3 3D printer frame; 2—PVC plastisol extrusion unit; 3—gear pump assembly; 4—pressure chamber with plastisol; 5—compressed air source; 6—cooling water circulation system; (**b**) PVC plastisol extrusion system: 1—PVC plastisol chamber; 2—gear pump assembly; 3—heat sink; 4—water jacket cooling; 5—cooling water circulation; 6—thermal barrier; 7—heater block assembly; and 8—nozzle.

**Figure 3 polymers-14-04683-f003:**
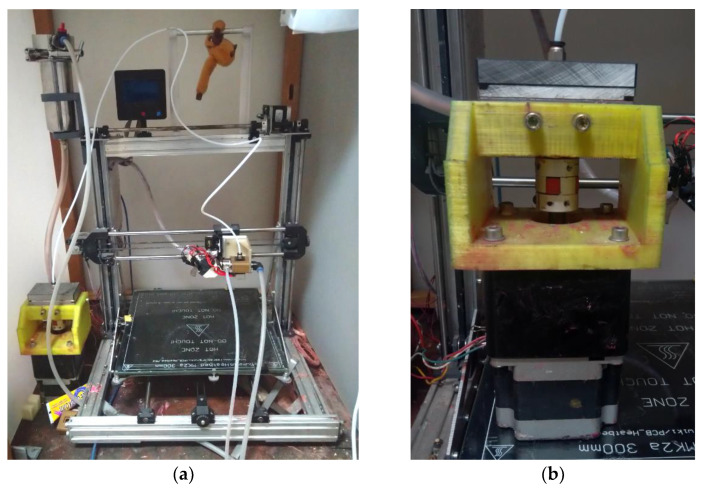
Picture of the actual AM setup: (**a**)—pump assembly; (**b**)—general view.

**Figure 4 polymers-14-04683-f004:**
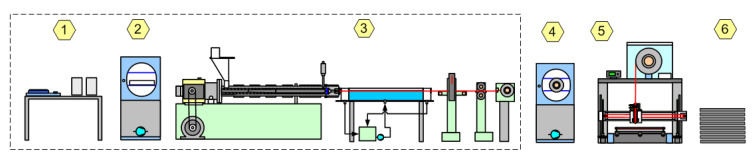
FFF AM from raw material to article: 1—raw material composition recipe; 2—composition pre-drying; 3—filament extrusion line; 4—filament pre-drying; 5—FFF AM process; and 6—article.

**Figure 5 polymers-14-04683-f005:**
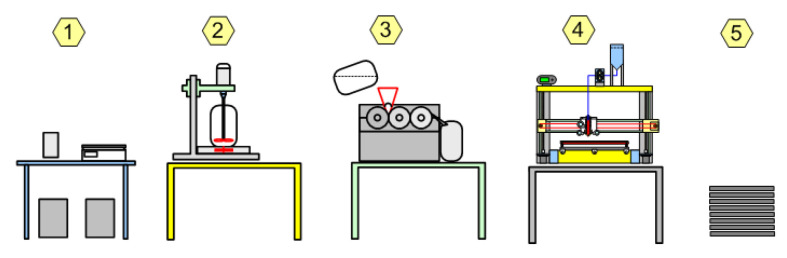
Plastisol AM from raw material to article: 1—composition formulation; 2—volumetric premix; 3—dispersion with rolling mill; 4—AM process; and 5—article.

**Figure 6 polymers-14-04683-f006:**
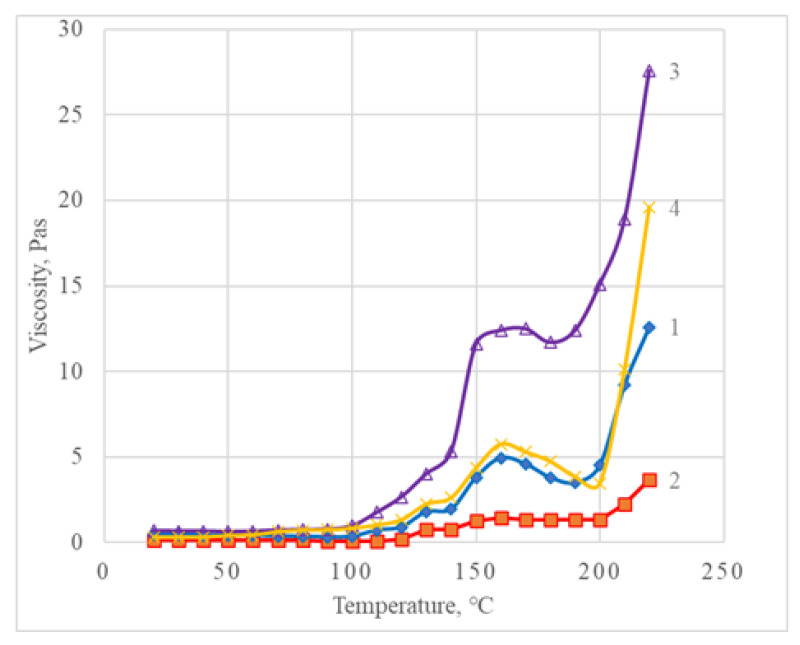
Dependence of viscosity on temperature for composition numbers from 1 to 4.

**Figure 7 polymers-14-04683-f007:**
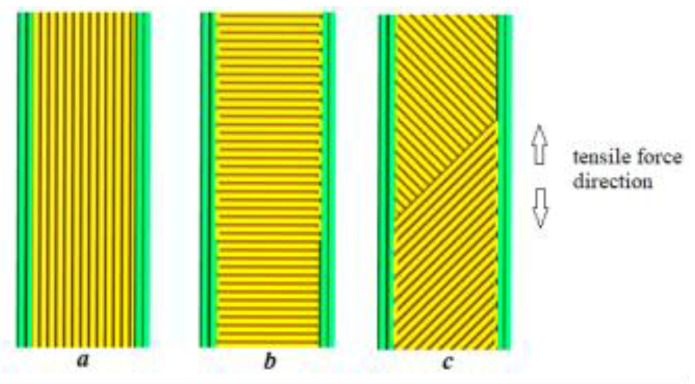
Layer orientation and direction of tensile force: (**a**)—90 deg; (**b**)—0 deg; and (**c**)—45 deg.

**Figure 8 polymers-14-04683-f008:**
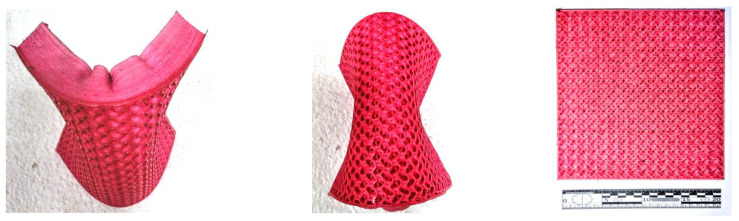
Open-cell AM material with shock-absorbing properties.

**Figure 9 polymers-14-04683-f009:**
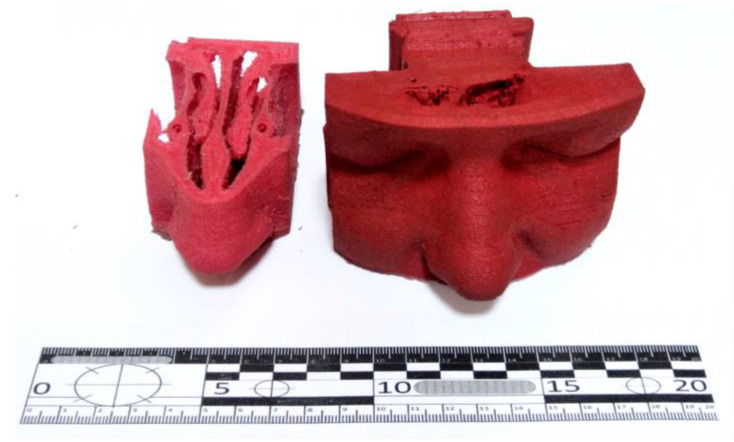
AM model for surgical training.

**Table 1 polymers-14-04683-t001:** Typical compositions used in the study.

Component	Composition Number					
	1	2	3	4	5	6
PVC	100	100	100	100	100	100
ESO	100	200	-	-	100	200
Stabilizer complex	2	2	4	4	2	2
DINP	-	-	50	100		
Filler	-	-	-	-	20	20

**Table 2 polymers-14-04683-t002:** Properties of soft PVC samples produced from plastisol compositions.

Properties	Composition Number					
	1	2	3	4	5	6
Melt flow rate, 190 °C	8	26	11	18	9	28
Density solid, g/cm^3^	1.08	1.05	1.11	1.07	1.12	1.09
Hardness Shore A	49	23	58	42	48	25
Tensile strength, MPa	7.1	4.3	9.4	7.5	8.2	4.6
Tensile elongation at break, %	270	430	210	305	280	440

**Table 3 polymers-14-04683-t003:** Typical AM parameters used in slicing.

Parameter	AM Process Parameter Set			
	AM1	AM2	AM3	AM4
Extrusion temperature, °C	180	180	190	190
Extrusion height, μm	200	200	200	200
Extrusion width, μm	500	500	500	500
Printing speed, mm/min	1500	1500	1500	1500
Platform temperature	80	80	80	80
Extrusion feed rate, mm^3^/m	80	160	80	160
Part air cooling, %	100	100	100	100
Nozzle diameter, μm	300	300	300	300
Retraction, mm	0	0	0	0
Top, wall, bottom layers	2	2	2	2
Infill type	Rectilinear	Rectilinear	Rectilinear	Rectilinear
Infill rate, %	100	100	100	100
Infill layer orientation, deg	45	45	45	45

**Table 4 polymers-14-04683-t004:** Properties of 3D printed PVC plastisol specimens under different AM conditions.

Composition	AM Process Parameter Set											
	AM1			AM2			AM3			AM4		
	Properties			Properties			Properties			Properties		
	σ	ε	SH	σ	ε	SH	σ	ε	SH	σ	ε	SH
1	6.3	180	48	5.8	190	46	6.7	170	48	6.6	190	47
2	3.9	260	22	4.1	240	23	4.2	290	22	4.1	300	21
3	7.2	200	59	6.9	210	57	7.4	220	58	6.7	210	57
4	4.1	270	40	4.3	315	41	4.3	310	42	4.4	310	42
5	6.2	190	49	5.9	200	48	6.6	170	50	6.4	160	49
6	4.0	240	24	3.8	230	24	4.1	250	25	3.8	280	24

σ—tensile strength, MPa; ε—tensile elongation at break, %; and SH—Shore hardness A scale.

**Table 5 polymers-14-04683-t005:** Influence of material flow orientation on specimen properties.

Composition	AM Process Layer Orientation					
	90 Deg		0 Deg		45 Deg	
	σ	ε	σ	ε	σ	ε
1	7.1	210	2.4	120	6.7	170
2	4.5	310	1.6	100	4.2	240
5	7.0	200	1.9	110	6.6	170
6	4.6	260	1.5	100	4.1	210

σ—tensile strength, MPa; ε—tensile elongation at break, %.

**Table 6 polymers-14-04683-t006:** Influence of infill density on material hardness.

Composition	Infill Density,%			
	100	90	80	70
	Shore Hardness A Scale			
1	48	46/45	41/37	38/37
2	22	20/19	18/17	17/16
5	50	48/48	45/41	41/40
6	25	23/22	20/17	18/18
Rectilinear pattern/Gyroid pattern				

## Data Availability

Not applicable.

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
