# Peer review of "New Approach for Extrusion Additive Manufacturing of Soft and Elastic Articles from Liquid-PVC-Based Consumable Materials"

_polymers, 2022, doi:10.3390/polym14214683_

Round 1

Reviewer 1 Report

polymers-1972990

New approach for extrusion additive manufacturing of soft and elastic articles from liquid PVC based consumable materials

This article develops an AM technique for deposition of liquid plastisol and showed better performance compared to conventional filament process. The way of writing is not clear and it is difficult for the readers to understand. The paper should be rewritten and proofread again thoroughly. The design constraints and modelling using developed AM setup is not clearly explained, without this the understanding the scientific novelty of this paper is difficult. What does soft and elastic articles mean in this paper?

The authors should clearly explain the novelty of this paper with scientific proof and results. Simply by doing the mechanical studies, hardness, density, MFR, viscosity and temperature measurement will not be sufficient for a high-quality technical publication. The authors should go through the below papers for better understanding and include it in the text.

https://doi.org/10.1016/j.addma.2017.06.009

https://doi.org/10.1002/9781119655053.ch11

https://doi.org/10.1007/s00170-015-7300-2

As such this paper will not be accepted for publication. However, the authors should consider these major modifications and rework it to improve the quality.

Author Response

Dear reviewer!

Thank you very much for the comments.

Design and origin of the developed AM process have been corrected and explained in more details.

The "soft and elastic articles" is used because the liquid form of the consumable is used. The liquid form of PVC plastisol contains an amount of plasticizers that can only be converted into articles with high softness and elasticity. Plastisol can be formulated with a lower plasticizer content, but it is not flowable and has typical paste properties.

The main objective of this paper is to show new AM possibilities for the application of liquid consumables in extrusion AM.

The process described in the article is capable of producing materials with low hardness, which is a known problem for conventional processes.

The scientific data presented aim to confirm the general possibility of obtaining acceptable material properties by these AM techniques. It is shown that the AM process can be carried out with composite materials and that it is easy to introduce additives and fillers. The complete formulation is listed for easy reproduction of the actual process.

The recommended papers are studied in detail and included in the reference list.

The current study is a part of our research work to develop AM foam material and articles with specific properties. It has military application as wave absorption material with flame retardant properties.

Reviewer 2 Report

Vozniak and coworkers reported an interesting work developing a novel additive manufacturing using a liquid consumable based on polyvinyl chloride plastisol. The work is novel and well-organized. However, some revisions in introduction and experiments are needed, the following questions must be addressed before publication.

1. The introduction about additive manufacturing should be further expanded in order to give readers a bigger picture. For example, there are many methods to print polymer, such as DLP and SLA method, which should be briefly introduced, and the following related paper must be cited (https://doi.org/10.1039/D1PY00705J; https://doi.org/10.1021/acsapm.2c00500). The advantages of Extrusion AM better than DLP and SLA can also be discussed.

2. Figure 4 an Figure 5, a scale bar could be added to the figures if it is possible, so this data is better presented.

3. How did authors get Figure 6? Is this a SEM data? Or optical image? This should be better noted in the figure caption and main text.

4.Figure 3, why don’t authors measure the sample 5 and 6? Would there be any differences? The authors can add some discussion.

Author Response

Dear reviewer!

Thank you very much for the comments.

The introduction has been expanded. A related paper has been cited.

Figure 4 an Figure 5, a scale bar could be added to the figures if it is possible, so this data is better presented.

Scale bars have been added.

How did authors get Figure 6? Is this a SEM data? Or optical image? This should be better noted in the figure caption and main text.

The figure is an optical image in polarized light. The author decides to remove this figure as it serves  no informative purpose. The description in the text is an indication that the surface of the article is different from that produced by conventional AM methods.

Figure 3, why don’t authors measure the sample 5 and 6? Would there be any differences? The authors can add some discussion.

The purpose of this part of the study is to describe the influence of layer orientation and filler content. Compositions 5 and 6 contain mineral fillers and indicate the possibility to formulate composites with the developed process.

Reviewer 3 Report

This is a very interesting and well written report on a new method of 3D printing with PVC. The article contains very little information about the chemistry of the polymers or their characterization, but the technical part about the new 3D printing technique provides enough very interesting information to warrant publication.

Author Response

Dear reviewer!

Thank you very much for the comments.

The publication contains complete formulation information for PVC plastisol material for easy reproduction. Since the thermomechanical transformation of plastisol is a physical rather than a chemical process, no additional chemical characterization is required. Thermal decomposition is a chemical process during the PVC AM process, which is described in the text as a technological factor affecting overall performance.

Round 2

Reviewer 1 Report

I have gone through the revised manuscript and completely satisfied with the revision. The authors have carefully addressed all the comments raised. Now, the papers technical depth is very much appropriate for the general knowledgeable individual working in the similar field. The results and discussions are acceptable and well-presented. Therefore, I accept the paper in its present form.

Reviewer 2 Report

The authors answered all my questions well.

The current version is good to publish.